# Proliferative Classification of Intracranially Injected HER2-positive Breast Cancer Cell Lines

**DOI:** 10.3390/cancers12071811

**Published:** 2020-07-06

**Authors:** Yuka Kuroiwa, Jun Nakayama, Chihiro Adachi, Takafumi Inoue, Shinya Watanabe, Kentaro Semba

**Affiliations:** 1Department of Life Science and Medical Bioscience, School of Advanced Science and Engineering, Waseda University, TWIns 2-2 Wakamatsu-cho, Shinjuku-ku, Tokyo 162-8480, Japan; css124244@akane.waseda.jp (Y.K.); adachi_c@asagi.waseda.jp (C.A.); inoue.t@waseda.jp (T.I.); ksemba@waseda.jp (K.S.); 2Department of Biomolecular Profiling, Translational Research Center, Fukushima Medical University, Hikarigaoka, Fukushima 960-1295, Japan; swata@mvc.biglobe.ne.jp; 3Department of Cell Factory, Translational Research Center, Fukushima Medical University, Hikarigaoka, Fukushima 960-1295, Japan

**Keywords:** HER2-positive breast cancer, brain metastasis, intracranial injection, in vivo imaging, microarray

## Abstract

HER2 is overexpressed in 25–30% of breast cancers, and approximately 30% of HER2-positive breast cancers metastasize to the brain. Although the incidence of brain metastasis in HER2-positive breast cancer is high, previous studies have been mainly based on cell lines of the triple-negative subtype, and the molecular mechanisms of brain metastasis in HER2-positive breast cancer are unclear. In the present study, we performed intracranial injection using nine HER2-positive breast cancer cell lines to evaluate their proliferative activity in brain tissue. Our results show that UACC-893 and MDA-MB-453 cells rapidly proliferated in the brain parenchyma, while the other seven cell lines moderately or slowly proliferated. Among these nine cell lines, the proliferative activity in brain tissue was not correlated with either the HER2 level or the HER2 phosphorylation status. To extract signature genes associated with brain colonization, we conducted microarray analysis and found that these two cell lines shared 138 gene expression patterns. Moreover, some of these genes were correlated with poor prognosis in HER2-positive breast cancer patients. Our findings might be helpful for further studying brain metastasis in HER2-positive breast cancer.

## 1. Introduction

Breast cancer cells metastasize to various organs, including the lung, bone, liver and brain. Breast cancer is classified into several subtypes based on the expression levels of hormone receptors and human epidermal growth factor receptor type 2 (HER2) [1]. HER2 overexpression is observed in about 25–30% of breast cancers [2,3], and approximately 30% of HER2-positive breast cancer patients are affected by brain metastases [4,5]. HER2-positive breast cancer has a high incidence of brain metastasis along with triple-negative breast cancer (TNBC) [6,7,8], but its molecular mechanisms are not well understood. Currently, radiation therapy and the resection of metastatic lesions are the first-line treatments for brain metastasis in HER2-positive breast cancer [9,10], while the chemotherapy and HER2-targeted therapy used for primary tumors are not actively recommended for brain metastasis. However, recent studies suggested that therapeutic drugs permeate through the damaged blood–brain barrier (BBB) after brain metastasis occurs and that they can inhibit brain metastasis progression to some extent [11,12,13,14,15]. These facts indicate the feasibility of developing chemotherapeutics and molecular targeted drugs that are effective for brain metastasis. To achieve the goal of developing such drugs, understanding the molecular mechanisms of tumor growth in the brain parenchyma is required.

To establish brain metastasis in experimental mouse models, the intracardiac injection method is generally used because the invasion-metastasis cascade from extravasation into the brain parenchyma to colonization can be mimicked by this method. However, this method induces systemic metastases, and the dispersion of tumor growth produced by this method is large [16], which is not appropriate for evaluating tumor growth only in the brain parenchyma. The intracranial injection method has been used in experimental mouse models of brain tumors [17]. Since this method does not induce metastasis to any other organs but to the central nervous system and the same number of tumor cells are injected into the same brain region with this method, we employed the intracranial injection method to evaluate the brain-colonizing ability of HER2-positive breast cancer cells to elucidate the mechanism of the later stages of brain metastasis.

In previous research, the TNBC cell line MDA-MB-231 and its brain metastatic derivatives have been widely used in experimental models of metastasis in breast cancer [18,19,20,21]. On the other hand, little is known about the mechanism of brain metastasis in HER2-positive breast cancer. Many of the existing studies on brain metastasis in HER2-positive breast cancer use a limited number of cell lines, such as BT-474 or MDA-MB-361 cells [21,22,23]. According to the previous studies, breast cancer cell lines SK-BR-3 and MCF-7 cells are sampled from pleural effusion, but they show weak metastatic activities to the lung [24,25]. In addition, MCF-7 cells were reported to have bone-colonizing potential [26], even though this activity is weak. Another study showed that DU4475 cells, which derive from the patient’s metastatic site, have a poor metastatic activity when injected into the murine left cardiac ventricle [27]. These reports suggested that cell lines sampled at the metastasis sites of the patients do not necessarily represent original metastatic potential. Conversely, some cell lines derived from primary sites have the potential to metastasize to other organs. For example, breast cancer cell lines BT-474 and HCC-1954 cells are sampled from the mammary gland, but they can survive in the brain [21,22,23]. Thus, it is important to reclassify the metastatic activities of cancer cell lines by transplantation models.

In this study, we evaluated the proliferative ability of nine HER2-positive breast cancer cell lines administered into the brain by intracranial injection and classified them into two groups based on the growth rate in brain tissue. Neither the HER2 level nor the HER2 phosphorylation status had a correlation with the growth rate in the brain parenchyma. Then, we performed microarray analysis of the nine HER2-positive breast cancer cell lines and identified signature genes associated with brain colonization.

## 2. Results

### 2.1. Identification of Two HER2-Positive Breast Cancer Cell Lines with Proliferative Ability in the Brain Parenchyma

We established nine HER2-positive breast cancer cell lines expressing the *luc2* gene—UACC-893, MDA-MB-453, HCC-2218, BT-474, ZR-75-1, UACC-812, MDA-MB-361, HCC-202, and HCC-1419 cells—with lentiviral vectors (Appendix A), and these cell lines were intracranially injected into NOD-SCID mice (HCC-1419 and HCC-2218 cells, n = 4; other cell lines, n = 3). All these cell lines originated from breast tissue, although some of them were sampled from metastasis sites of the subject [28,29,30,31,32,33] (Table 1). In addition, cell morphology differed by cell line (Appendix A). Although all the cell lines expressed HER2, the patterns of hormone receptor expression and cancer-related gene expression, HER2 expression level, and mutation profile of tumor suppressor genes vary by cell line (American Type Culture Collection (ATCC); Cancer Cell Line Encyclopedia (CCLE)) [34,35,36,37,38,39,40,41,42,43,44,45,46,47,48,49] (Table 2 and Appendix A). Among these nine cell lines, seven had *tumor protein p53* (*TP53)* mutations and five had *phosphatidylinositol-4,5-bisphosphate 3-kinase catalytic subunit alpha* (*PIK3CA)* mutations. 

When injected into the murine brain parenchyma, UACC-893 and MDA-MB-453 cells proliferated rapidly by 28 days after injection (Figure 1A). The growth rate of ZR-75-1, UACC-812, MDA-MB-361, HCC-202, and HCC-1419 cells was slow (Figure 1B). HCC-2218 and BT-474 cells moderately proliferated (Figure 1B). Following these results, we classified these cell lines into two groups based on brain-colonizing potential. UACC-893 and MDA-MB-453 cells were categorized into the rapid growth group (RG), while HCC-2218, BT-474, ZR-75-1, UACC-812, MDA-MB-361, HCC-202, and HCC-1419 cells were categorized into the medium to slow growth group (MSG) (Figure 1A,B). Among these nine cell lines, the RG cell line MDA-MB-453 proliferated relatively faster in vitro, while UACC-893 cells, also categorized into the RG, proliferated relatively slower in vitro (Appendix A). Two MSG cell lines ZR-75-1 and BT-474 proliferated relatively faster in vitro, while the other five MSG cell lines grew relatively slower in vitro (Appendix A). These results show that as for the nines cell lines used in this study, the growth rate in vivo was not correlated with that in vitro (Figure 2A).

HER2 overexpression in breast cancer cells promotes brain colonization [50]. To examine the correlation of HER2 expression levels with brain colonization, we conducted Western blotting with the nine HER2-positive breast cancer cell lines. Although the RG cell line UACC-893 showed high expression levels of HER2, some MSG cell lines, such as HCC-2218, UACC-812, and HCC-1419, also overexpressed HER2 (Appendix A). We also examined the major autophosphorylation sites of HER2, Tyr1221/1222 and Tyr1248 [51,52], but among these nine cell lines, we could not observe any correlation between HER2 phosphorylation levels and brain-colonizing potential (Figure 2B). Consequently, considering previous studies on brain metastasis in breast cancer, the potential of brain colonization might be affected by HER2 to some extent [53], but the presence of novel mechanisms of cell proliferation in the brain parenchyma is also possible.

### 2.2. Growth Activities of The RG Cell Lines Induced by in Vitro Coculture with Primary Glial Cells

Metastatic cancer cells grow in the brain, interacting with brain-residential glial cells [54]. To characterize the RG cell lines in terms of interaction with glial cells, we cocultured UACC-893 and MDA-MB-453 cells with murine primary glial cells in serum-free medium. As a result, both UACC-893 and MDA-MB-453 cells were able to grow attached to the layer of glial cells, while they floated and grew as aggregates when cultured alone in the same medium (Figure 3). The MSG cell lines also grew under this coculture condition, suggesting that the in vivo condition is more stringent for these cell lines.

### 2.3. Gene Expression Analysis Between Two Groups of HER2-Positive Breast Cancer Cell Lines

To identify the genes responsible for regulating brain-colonizing potential in HER2-positive breast cancer, we conducted microarray analysis of the nine HER2-positive breast cancer cell lines. The microarray data obtained in the previous study were reanalyzed for the present study [55]. The expression level of each gene was converted to a z-score, and genes with a z-score > 1.0 and z-score < −1.0 were extracted as upregulated genes and downregulated genes, respectively (Figure 4A). To extract signature genes that were differentially expressed between the RG and MSG, we first calculated the log fold-change (FC) ratio of the RG average gene expression to the MSG average gene expression. Second, the logFC ratio of the expression in MDA-MB-453 cells to that in UACC-893 cells (both in the RG) was calculated. As a result, 57 upregulated genes and 81 downregulated genes were extracted and defined as brain-colonizing signature genes in HER2-positive breast cancer (Figure 4B). *Transmembrane 4 L-six family member 1* (*TM4SF1*) and *aspartate beta-hydroxylase* (*ASPH*), both reported as multiorgan metastasis-promoting genes in breast cancer [56,57], were included in these signature genes.

To examine the correlation between the expression level of the signature genes and the prognosis of HER2-positive breast cancer patients, we conducted survival analysis using the Molecular Taxonomy of Breast Cancer International Consortium (METABRIC) dataset (Table 3; Figure 5A,B). High expression of 11 genes (*CD9 molecule* (*CD9*), *ubiquitin-like 3* (*UBL3*), *transmembrane 4 L six family member 1* (*TM4SF1*), *microsomal glutathione S-transferase 3* (*MGST3*), *netrin 4* (*NTN4*), *NOVA alternative splicing regulator 1* (*NOVA1*), *aspartate beta-hydroxylase* (*ASPH*), *embryonic ectoderm development* (*EED*), *ubiquitin specific peptidase 53* (*USP53*), *mediator complex subunit 17* (*MED17*), and *sphingomyelin phosphodiesterase acid like 3B* (*SMPDL3B*)) and low expression of six genes (*transcription termination factor 2* (*TTF2*), *cystatin A* (*CSTA*), *ATP binding cassette subfamily C member 2* (*ABCC2*), *heat shock protein family B (small) member 8* (*HSPB8*), *flotillin 2* (*FLOT2*), and *NLR family pyrin domain containing 2* (*NLRP2*)) was correlated with poor prognosis of HER2-positive breast cancer patients.

## 3. Discussion

Brain metastasis worsens the prognosis and survival of breast cancer patients [15]. HER2-positive breast cancer has the shortest median time period between the diagnosis of breast cancer and the detection of brain metastasis among all breast cancer subtypes [15], yet its molecular mechanism has not been elucidated in detail.

Many studies have relied on BT-474 and MDA-MB-361 cells for brain metastasis models of HER2-positive breast cancer [21,22,23]. However, no studies in breast cancer brain metastasis have used multiple cell lines and compared gene expression or proliferative ability among them. In this study, we intracranially injected nine HER2-positive breast cancer cell lines and observed differences among these cell lines in growth rate. Moreover, we found that BT-474 and MDA-MB-361 cells proliferated relatively slowly in the brain parenchyma, suggesting that it might be more suitable to use UACC-893 or MDA-MB-453 cells for brain metastasis assays. However, there are more HER2-positive breast cancer cell lines besides those used in this study. It is possible that the novel RG cell lines will be identified by using the intracranial injection method. Although cell lines categorized into the MSG continued to proliferate with low speed until the endpoint of the experiment (56-84 days after injection; Appendix A), we regarded them as the MSG from the viewpoint of metastatic potential.

Both UACC-893 and MDA-MB-453 cells had the H1047R mutation in *PIK3CA*, while no MSG cell lines had that mutation. Instead, two MSG cell lines, MDA-MB-361 and HCC-202 cells, had the PIK3CA-E545K mutation in common. H1047R and E545K are major mutations in *PIK3CA* and are often found in cancer patients, including breast cancer patients [58,59]. Both H1047R and E545K are activating mutations, and H1047R is a stronger activating mutation than E545K, promoting the growth of cancer cells and angiogenesis [60]. *PIK3CA* encodes p110α, a subunit of phosphoinositide 3-kinase (PI3K), and the proliferation signal from PI3K is transduced to protein kinase B (PKB; AKT) [59,61,62]. In a previous study, the pan-AKT inhibitor GDC-0068 decreased the viability of MDA-MB-453 cells in vitro [63]. Considering that activation of the PI3K/AKT pathway is observed in breast cancer patients with brain metastasis [64,65], this signaling pathway might be a potential target for curing brain metastasis. On the other hand, the results of signaling analysis show that the P-AKT (S473) level did not correlate with growth activity in the brain parenchyma (Appendix A). A previous study demonstrated that in some cases, PI3K/AKT signal activation was observed only in the brain microenvironment and that inhibition of PI3K reduced the invasion ability of breast cancer cells induced by macrophages and microglia under coculture conditions [66]. Based on this fact, in brain-metastatic HER2-positive breast cancer cells, PI3K/AKT signaling might be activated in the in vivo brain microenvironment, or there might be other mechanisms for cell growth in the brain parenchyma. According to the mutation profiles from CCLE, 15 genes were found to be mutated both in UACC-893 and MDA-MB-453 cells but not in any MSG cell lines (Appendix A; in this study, mutations without protein change and mutations in splice sites were not regarded as gene mutations.). Aberrant expression of *interleukin 1 receptor associated kinase 1* (*IRAK1*), *serpin family I member 2* (*SERPINI2*), and *WW and C2 domain containing 1* (*WWC1*) was reported to be associated with breast cancer metastasis [67,68,69], although the relationship between their mutation status and brain metastasis remains to be elucidated. Additionally, both UACC-893 and MDA-MB-453 cells had frame shift insertion (A1733fs) in *listerin E3 ubiquitin protein ligase 1* (*LTN1*), but its effects on brain metastasis are unknown.

To characterize the RG cell lines, we cocultured nine HER2-positive breast cancer cell lines with glial cells for six days in serum-free medium. Although we confirmed that both UACC-893 and MDA-MB-453 cells proliferated well when cocultured with glial cells, not only the RG cells but also some MSG cells grew under these conditions (Figure 3), suggesting that our coculture system of tumor cells and glial cells only partially mimicked the brain microenvironment. We also found that cocultured HER2-positive breast cancer cells adhered to glial cells and proliferated by using them as a scaffold, although they grew as nonadherent cells when cultured without glial cells in this serum-free medium. According to previous studies using MDA-MB-231 cells, continuous direct contact between tumor cells and astrocytes upregulates genes related to survival, chemoresistance, and growth in tumor cells [70,71]. Considering these reports, our coculture system could be employed for measuring the direct interaction between tumor cells and primary glial cells.

Based on the microarray analysis results, we believed there are novel mechanisms of cell proliferation in addition to HER2 signaling in brain tissue. To confirm the contribution of HER2 to brain colonization, we examined HER2 expression in tumor cells that survived in brain tissue after intracranial injection (Appendix A). We also examined the phosphorylation levels of P-HER2 (Tyr1221/1222) and P-HER2 (Tyr1248), which are the binding sites of SHC and activate RAS/MAPK signaling [72]. The HER2 expression level and its phosphorylation status were almost the same between the original cell lines and surviving cells. A previous study examining the roles of HER2 in brain metastasis demonstrated that HER2 promotes the growth of cancer cells in the brain [50], but it does not exclude the possibility of a HER2-independent brain-colonizing mechanism. These facts suggest that certain expressions of HER2 might be required for tumor growth in the brain parenchyma, although it is not the only factor that mediates brain metastasis progression. To identify novel genes that promote brain colonization, we conducted microarray analysis and extracted signature genes from the RG cell lines. According to the survival analysis, 11 upregulated genes and six downregulated genes, representing the brain-colonizing signature, were correlated with the prognosis of HER2-positive breast cancer patients. These genes might have the potential to be novel prognostic markers for HER2-positive breast cancer patients. *TM4SF1*, one of the upregulated signature genes, is a known multiorgan metastasis-promoting gene, and knockdown of *TM4SF1* significantly reduces brain metastasis in mouse mammary tumor cells transformed with rat erb-b2 receptor tyrosine kinase 2 (Erbb2) [56,73]. Overexpression of *CD9*, another upregulated signature gene, promotes bone metastasis in TNBC [74]. *CD9* is also known as a marker of extracellular vesicles (EVs), and EVs are associated with breast cancer metastasis [75]. Considering that treatment with anti-CD9 antibodies decreases metastasis to the lungs, lymph nodes, and thoracic cavity in TNBC [75], *CD9* might also be a potential target of brain metastasis treatment in HER2-positive breast cancer. *ASPH*, which encodes aspartate beta-hydroxylase, was also highly expressed in the RG cell lines. When injected into the mouse mammary fat pad, ASPH-overexpressing MDA-MB-231 cells metastasize to the lungs, lymph nodes, spleen, intestine, mesentery, and liver by activating the Notch signaling pathway and subsequent synthesis and release of metastasis-inducible exosomes [57]. In addition, the inhibition of ASPH suppresses the migration of ASPH-overexpressing MDA-MB-231 cells in vitro [57], suggesting that ASPH might be a potential target for the treatment of brain metastasis in HER2-positive breast cancer as well. High expression of *MGST3* and *USP53* was also correlated with poor clinical outcome in HER2-positive breast cancer, although the contribution of these two genes to brain metastasis has not been reported. Among the downregulated signature genes, low expression of *CSTA* and *HSPB8* was correlated with unfavorable outcomes in HER2-positive breast cancer, while no correlation was observed with outcome in all breast cancers. On the other hand, low expression of *TTF2*, *ABCC2*, *FLOT2*, and *NLRP2* was correlated with poor prognosis in HER2-positive breast cancer patients, whereas their low expression was correlated with better prognosis in the overall METABRIC cohort. This result implies that some candidate prognostic marker genes applicable in breast cancer overall work inversely in HER2-positive breast cancer and that the results of genetic testing should be interpreted based on patient subtype.

## 4. Materials and Methods

### 4.1. Cell Culture

MDA-MB-453, UACC-893, HCC-2218, HCC-1419 (ATCC, Manassas, VA, USA), and ZR-75-1 cells (Institute of Development, Aging and Cancer (IDAC), Miyagi, Japan) were cultured in Roswell Park Memorial Institute medium (RPMI-1640, FUJIFILM Wako Pure Chemical Corporation, Osaka, Japan) supplemented with 10% heat-inactivated fetal bovine serum (FBS, NICHIREI BIOSCIENCES INC., Tokyo, Japan), 100 U/mL penicillin (Meiji-Seika Pharma Co., Ltd., Tokyo, Japan), and 100 µg/mL streptomycin (Meiji-Seika Pharma) at 37 °C with 5% CO_2_. MDA-MB-361 and HCC-202 cells (ATCC) were cultured in RPMI-1640 (FUJIFILM Wako Pure Chemical Corporation) supplemented with 15% heat-inactivated FBS, 100 U/mL penicillin (Meiji Seika Pharma), and 100 µg/mL streptomycin (Meiji-Seika Pharma) at 37°C with 5% CO_2_. BT-474, UACC-812, and 293T cells (ATCC) were cultured in Dulbecco’s modified Eagle’s medium (DMEM, FUJIFILM Wako Pure Chemical Corporation) supplemented with 10% heat-inactivated FBS, 100 U/mL penicillin (Meiji Seika Pharma), and 100 µg/mL streptomycin (Meiji-Seika Pharma) at 37 °C with 5% CO_2_. Luc2-expressing breast cancer cell lines were established by infection with lentivirus (pLenti-P_EF1_-luc2-IRES-Bla^R^) and selection by blasticidin (Kaken Pharmaceuticals Co. Ltd., Tokyo, Japan).

### 4.2. Viral Infection

pPACKHI-REV, pPACKHI-GAG, pVSV-G, and pLenti-P_EF1_-luc2-IRES-Bla^R^ were cotransfected into 293T cells by the calcium phosphate method. Two days later, the culture supernatant was collected for lentivirus infection. UACC-893, MDA-MB-453, HCC-2218, BT-474, ZR-75-1, UACC-812, MDA-MB-361, HCC-202, and HCC-1419 cells seeded in 12-well plates were incubated in two-fold diluted lentiviral solution for 24 h, and then, the media were replaced with fresh culture media.

### 4.3. Luciferase Assay

A total of 1 × 10^5^ cells from luc2-introduced HER2-positive breast cancer cell lines were lysed in 200 µL Cell Culture Lysis Reagent (CCLR) (25 mM Tris-phosphate, 2 mM Na_2_EDTA, 10% glycerol, 1% Triton X-100, and 2 mM DTT). The cell lysate was centrifuged, and 10 µL supernatant was suspended in 90 µL Milli-Q. Twenty microliters of the 10-fold diluted lysate was added to 3 wells of a white 96-well plate. Fifty microliters of firefly solution (7 mM DTT, 280 µM CoA, 210 µM ATP, 196.5 µg/mL luciferin, and 41% Firefly Salts (Firefly Salts: 150 mM Tris-HCl, 75 mM NaCl, and 3 mM MgCl_2_)) was added to each well, and bioluminescence was measured using TriStar^2^ S LB942 (Berthold Technologies GmbH & Co. KG, Bad Wildbad, Germany).

### 4.4. Western Blotting

Nine HER2-positive breast cancer cell lines expressing the *luc2* gene were seeded in 6 cm dishes. Cells were lysed in 350 µL RIPA buffer (50 mM Tris-HCl (pH 8.0), 150 mM NaCl, 1% NP-40, 0.1% SDS, 1 mM PMSF, 1 mM Na_3_VO_4_, and 10 mM NaF). Protein concentrations of cell lysates were measured by the BCA method. Then, 150 µL 3× SDS sample buffer (150 mM Tris-HCl (pH 6.8), 30% glycerol, 0.015% BPB, and 6% SDS) was added to each cell lysate (300 µL). Protein samples (15 µg) were subjected to SDS-PAGE and transferred to poly vinylidene di-fluoride (PVDF) membranes (Immobilon-P; Millipore, Darmstadt, Germany). Blots were incubated overnight with primary antibodies (targeting P-ERBB2 (Tyr1221/1222), 2243S; P-ERBB2 (Tyr1248), 2247S; ERBB2, 4290S; P-STAT3 (Tyr705), 9145S; STAT3, 9139S; P-AKT (Ser473), 4060S; AKT, 4691S; P-ERK (Thr202/Tyr204), 4370S; ERK, 4695S; and β-ACTIN, sc-69879) and for 1 h with peroxidase-linked secondary antibody (Anti-rabbit IgG, HRP-linked antibody, GE Healthcare Life Sciences; Anti-mouse IgG, HRP-linked antibody, GE Healthcare Life Sciences). Proteins were detected using chemiluminescent HRP substrate (Immobilon^TM^ Western; Millipore, Darmstadt, Germany).

### 4.5. Growth Analysis of HER2-Positive Breast Cancer Cell Lines in Vitro

A total of 1.5 × 10^5^ cells from each HER2-positive breast cancer cell line were plated in 12-well plates (n = 3 for each time point). Cells were harvested using 0.25% trypsin every other day, and the cell number was counted.

### 4.6. Mouse Xenograft and In Vivo Bioluminescence Imaging

Six-week-old female NOD.CB-17-Prkdcscid/J mice (NOD-SCID; Charles River Laboratories Japan, Inc., Kanagawa, Japan) were used for intracranial transplantation. A total of 1.0 × 10^5^ luc2-expressing breast cancer cell lines suspended in 4 µL D-PBS (−) (FUJIFILM Wako Pure Chemical Corporation) were injected intracranially using a 26-gauge syringe. After intracranial transplantation, 200 µL D-luciferin (15 mg/mL) (Gold Biotechnology, Inc., St. Louis, MO, USA) was injected intraperitoneally into NOD-SCID mice, and bioluminescence imaging was conducted every seven days using an in vivo imaging system (IVIS Lumina XRMS; Perkin-Elmer, Waltham, MA, USA). All animal experiments were approved by the Animal Committee of Waseda University.

### 4.7. Isolation of Tumor Cells Surviving in Brain Tissue

Mouse brains were extracted 4–12 weeks after intracranial transplantation. Each brain was cut into small pieces in 5 mL culture medium in four 6 cm dishes. Tumor cells that survived in the brain parenchyma were isolated and cultured as previously described [76]. After cells proliferated and reached approximately 60% confluence, drug-induced selection was conducted using blasticidin (Kaken Pharmaceutical Co. Ltd.) for four days to isolate tumor cells from brain-resident cells.

### 4.8. Coculture Assay with Mouse Primary Glial Cells and HER2-Positive Breast Cancer Cell Lines

Primary cortical glial cultures were prepared from E17 mice as described previously [77,78] with modifications: dissociated cerebral cortex hemispheres were plated in PEI-coated 10 cm^2^ culture dishes with Neurobasal-A medium (Gibco/Thermo Fisher Scientific, Waltham, MA, USA) supplemented with L-glutamine, MACS Neurobrew-21 (Miltenyi Biotec B.V. & Co. KG, Bergisch Gladbach, Germany), and 0.05% penicillin-streptomycin. After 10 days in culture, the cells were suspended with 0.05% trypsin in Hanks’ balanced salt solution without Ca^2+^ and Mg^2+^ and plated on PEI-coated 24-well plates with the same culture medium (5 × 10^4^ cells/well). On the following day, 1.5 × 10^4^ cells of each of the nine HER2-positive breast cancer cell lines were seeded on glial cells. These cells were incubated for 6 days at 37 °C in a 5% CO_2_ incubator. Pictures were taken every other day.

### 4.9. DNA Microarray Analysis and Statistical Data Analysis

DNA microarray data provided in the previous research were subjected to reanalysis to study the brain colonization of HER2-positive breast cancer cell lines [55]. Gene probes that met the following criteria were extracted and analyzed further with Venn diagrams: “–1.0 < RG logFC < 1.0 and RG z-score > 1.0”, “–1.0 < RG logFC < 1.0 and RG z-score < –1.0”, “MSG z-score > 1.0”, and “MSG z-score < –1.0”. The Venn diagrams were created using ‘GeneVenn’ (http://genevenn.sourceforge.net/). To construct the heatmap, gene probes with the following criteria were extracted: “–1.0 < RG logFC < 1.0, RG-MSG logFC > 1.0, and RG z-score > 1.0” or “–1.0 < RG logFC < 1.0, RG-MSG logFC < –1.0, and RG z-score < –1.0”. The heatmap was drawn by the ‘pheatmap’ package using the R statistical programming language (version 3.6.1., https://cran.r-project.org/bin/windows/base/old/3.6.1/).

### 4.10. Survival Analysis

Survival analysis was conducted for breast cancer patients in the Molecular Taxonomy of Breast Cancer International Consortium (METABRIC) datasets as previously described [26,79]. We used a cohort of HER2-positive breast cancer patients and all breast cancer patients for this survival analysis.

## 5. Conclusions

In conclusion, we measured the brain-colonizing potential of nine HER2-positive breast cancer cell lines by intracranial injection and identified two cell lines that rapidly proliferated in the brain parenchyma. Some of the brain-colonizing signature genes extracted from these cell lines have the potential to be utilized as prognostic marker genes for HER2-positive breast cancer patients. Our findings might contribute to the further understanding of the mechanisms of brain metastasis in HER2-positive breast cancer.

## Figures and Tables

**Figure 1 cancers-12-01811-f001:**
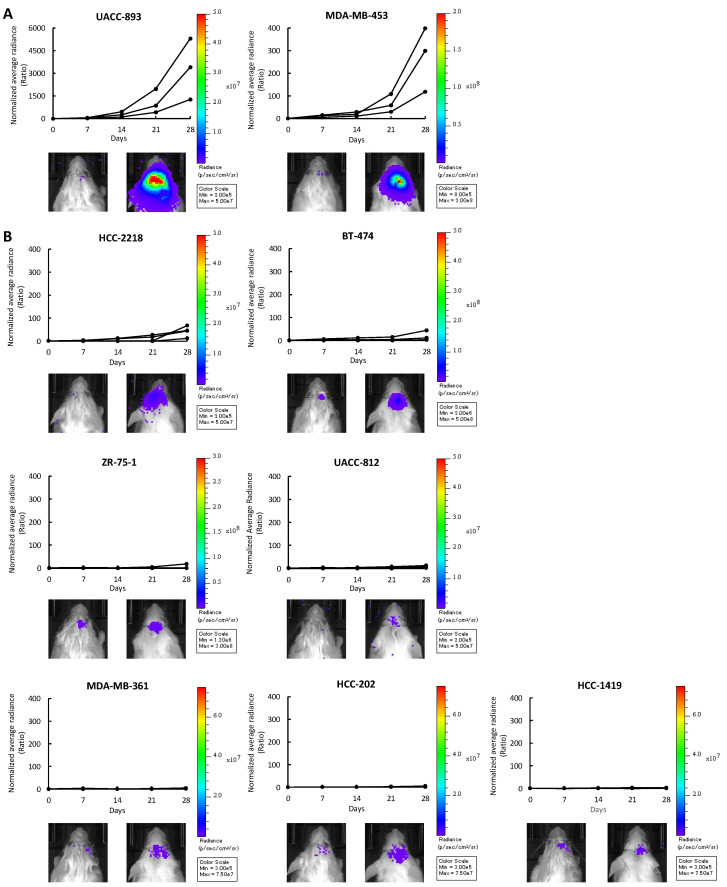
Intracranial injection of nine HER2-positive breast cancer cell lines. UACC-893-luc2, MDA-MB-453-luc2, HCC-2218-luc2, BT-474-luc2, ZR-75-1-luc2, UACC-812-luc2, MDA-MB-361-luc2, HCC-202-luc2, and HCC-1419-luc2 cells were intracranially injected into NOD-SCID mice (HCC-2218-luc2 and HCC-1419-luc2, n = 4; others, n = 3). Cell proliferation was quantified by measuring bioluminescence every seven days and plotted in the form of a growth ratio. Each line shows the corresponding mouse. Left: Bioluminescence on day 0. Right: Bioluminescence on day 28. (**A**) Cell lines in the rapid growth group (RG). (**B**) Cell lines in the moderate to slow growth group (MSG).

**Figure 2 cancers-12-01811-f002:**
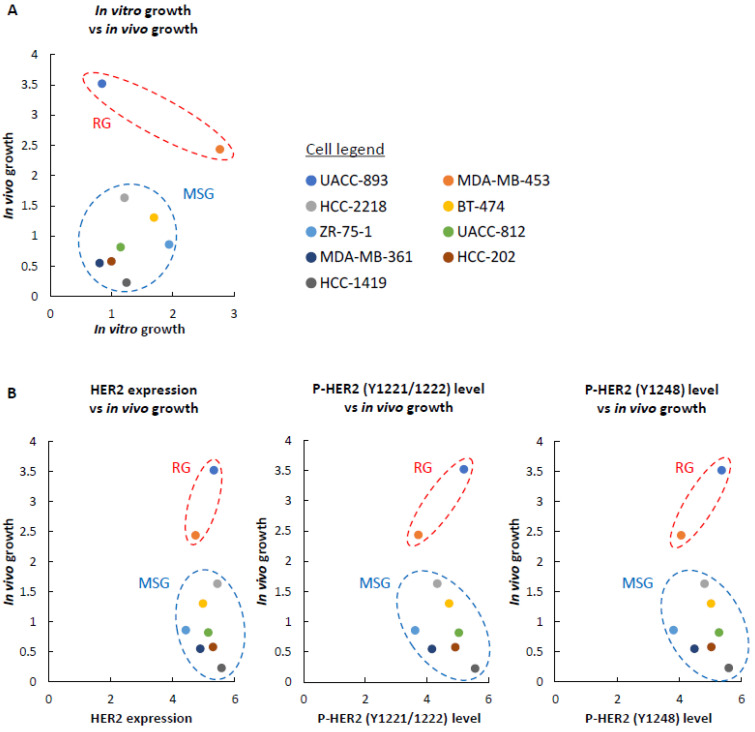
The relationship between in vivo growth and in vitro growth, in vivo growth and HER2 profiles. In all graphs, the vertical axis represents in vivo growth, which is presented on a log10 scale (mean normalized average radiance (ratio) on day 28). (**A**) The relationship between brain-colonizing ability and proliferative activity in vitro. The horizontal axis represents in vitro growth. The cell number on day 4 was converted to a log2 (N(4)/N_0_) value for three replicates and their mean value was defined as in vitro growth. N(4) = The cell number on day 4. N_0_ = The number of cells seeded on day 0 (=1.5 × 10^5^ cells). (**B**) The contribution of HER2 expression and HER2 phosphorylation to growth activity in vivo. The horizontal axis represents HER2 expression, P-HER2 (Y1221/1222) level, and P-HER2 (Y1248) level. Each band intensity in Appendix A was quantified as a raw integrated density (RawIntDen) using ImageJ. The RawIntDen measured by ImageJ was converted to a log10 (RawIntDen) value for HER2 expression, the P-HER2 (Y1221/1222) level, and the P-HER2 (Y1248) level.

**Figure 3 cancers-12-01811-f003:**
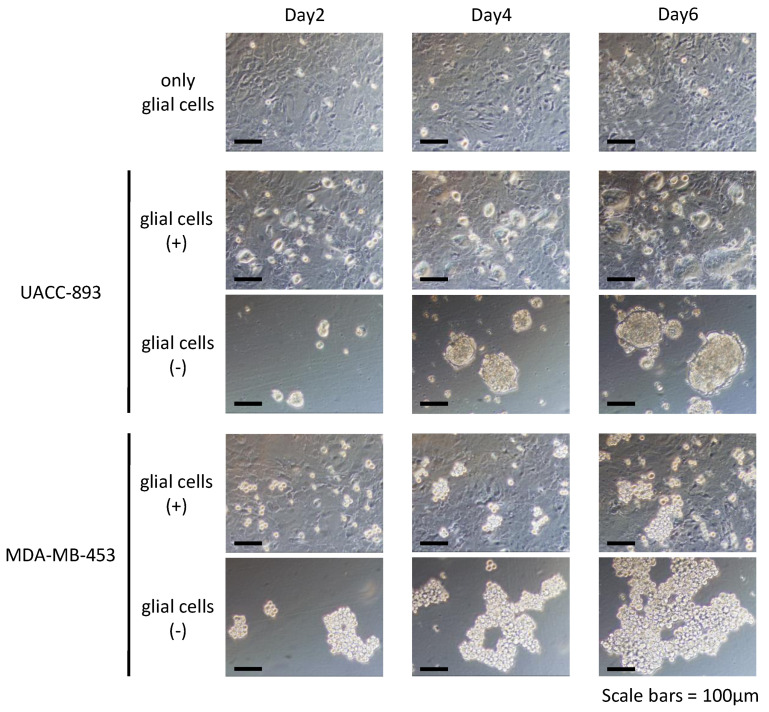
Coculture of HER2-positive breast cancer cell lines and murine glial cells. Pictures of two RG cell lines cultured in serum-free medium with or without glial cells for six days are shown (n = 3). Pictures were taken every other day. Scale bars = 100 µm.

**Figure 4 cancers-12-01811-f004:**
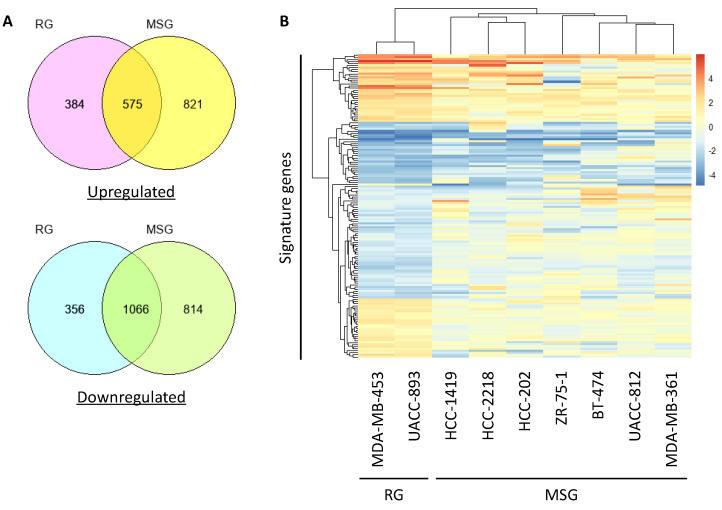
Signature genes associated with brain colonization in the RG. (**A**) The number of upregulated genes and downregulated genes in the RG or MSG is described in Venn diagrams. (**B**) Genes that were differentially expressed between the RG and MSG according to microarray data were analyzed by hierarchical clustering and are shown in a heatmap.

**Figure 5 cancers-12-01811-f005:**
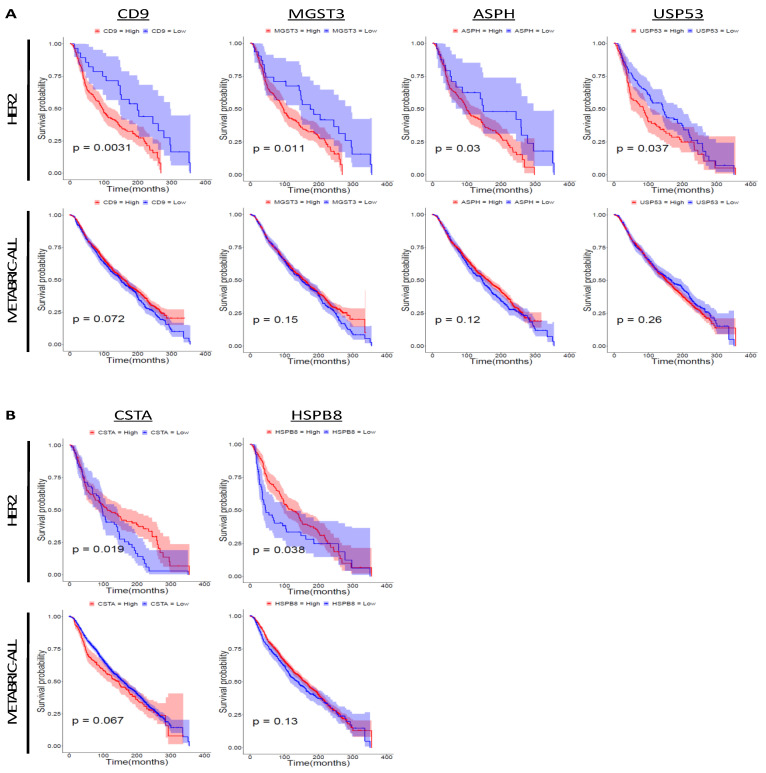
Survival analysis of signature genes upregulated in the RG. Survival analysis was conducted using the METABRIC dataset. The colored region along the curve shows the 95% confidential interval. Nine upregulated genes (*kynurenine 3-monooxygenase* (*KMO*), *inhibin subunit beta B* (*INHBB*), *zinc finger protein 663, pseudogene* (*ZNF663P*), *egl-9 family hypoxia-inducible factor 3* (*EGLN3*), *transmembrane protein 45B* (*TMEM45B*), *MAGE family member F1* (*MAGEF1*), *COMM domain containing 2* (*COMMD2*), *ZNF205 antisense RNA 1* (*ZNF205-AS1*), and *multiple EGF like domains 6* (*MEGF6*)) and four downregulated genes (*DNAJC25-GNG10 readthrough* (*DNAJC25-GNG10*), *intraflagellar transport 57* (*IFT57*), *MYB proto-oncogene like 1* (*MYBL1*), and *adhesion molecule with Ig like domain 2* (*AMIGO2*)) were not analyzed because there were no data for these genes in the METABRIC dataset. (**A**) Survival analysis of signature genes upregulated in the RG. The number of specimens was as follows: METABRIC-ALL: *CD9* (high = 1369, low = 535), *MGST3* (high = 1423, low = 481), *ASPH* (high = 1432, low = 472), and *USP53* (high = 1404, low = 500); HER2-positive: *CD9* (high = 192, low = 28), *MGST3* (high = 189, low = 31), *ASPH* (high = 196, low = 24), and *USP53* (high = 113, low = 107). (**B**) Survival analysis of signature genes downregulated in the RG. The number of specimens was as follows: METABRIC-ALL: *CSTA* (high = 286, low = 1618) and *HSPB8* (high = 1507, low = 397); HER2-positive: *CSTA* (high = 150, low = 70) and *HSPB8* (high = 167, low = 53).

**Table 1 cancers-12-01811-t001:** Characteristics of Nine HER2-positive Breast Cancer Cell Lines.

Ell Line	Metastasis Sites in The Subject	Sampled Site	Cell Morphology	Culture Properties	References
UACC-893	Lymph nodes	Primary breast	Epithelial	Adherent	ATCC [29]
MDA-MB-453	Nodes, Brain, Pleural and Pericardial cavities	Pericardial effusion (metastasis site)	Rounded	Adherent	ATCC [30]
HCC-2218	Lymph nodes	Primary breast	Rounded	Non-adherent	ATCC [31]
BT-474	N/A	Primary breast	Epithelial	Adherent	ATCC [32]
ZR-75-1	Ascites	Ascitic effusion (metastasis site)	Epithelial	Adherent	ATCC [33]
UACC-812	Neck, Liver	Primary breast	Epithelial	Adherent	ATCC [29]
MDA-MB-361	Brain	Brain (metastasis site)	Epithelial	Adherent	ATCC [30]
HCC-202	Lymph nodes	Primary breast	Epithelial	Adherent	ATCC [31]
HCC-1419	Lymph nodes	Primary breast	Epithelial	Adherent	ATCC [31]

N/A: Not available. Morphology was judged by Appendix A and categorized based on [28].

**Table 2 cancers-12-01811-t002:** Gene Mutation Profile of Nine HER2-positive Breast Cancer Cell Lines.

Cell line	BRCA1	BRCA2	BRAF	HRAS	PIK3CA	TP53	PTEN	References
UACC-893	WT	WT	WT	WT	H1047R	R342 * ¶	WT	CCLE [34,35,42,43,44,45,46,47]
MDA-MB-453	WT	WT	WT	WT	H1047R	Deletion(30bp) at codon 367 ¶ §	E307K	CCLE [34,35,42,43,44,45,46,47,48]
HCC-2218	WT	WT	E296K	N/A	WT	R283C §	WT	CCLE [46,47]
BT-474	WT ‡	S3094 *	WT	WT	K111N	E285K	WT	CCLE [34,35,42,43,44,45,46,47]
ZR-75-1	WT	WT	WT	E162K §	WT	WT	L108R	CCLE [34,35,42,44,45,46,47]
UACC-812	WT ‡	WT	WT	WT	WT	WT	WT	CCLE [34,35,44,45,46,47,48]
MDA-MB-361	WT ‡	N1657S	V600E §	WT	E545K/K567R	E56 */E166 * §	WT	CCLE [34,35,42,43,44,45,46,47,48]
HCC-202	WT	WT	N/A	N/A	E545K/L866F	T284fs	WT	CCLE [46]
HCC-1419	WT	WT	N/A	R128W	WT	Y220C/APA74fs	WT	CCLE [43,46]

N/A: Not available, WT: wild-type, BRCA1: BRCA1 DNA repair associated, BRCA2: BRCA2 DNA repair associated, BRAF: B-Raf proto-oncogene, serine/threonine kinase, HRAS: HRas proto-oncogene, GTPase, PIK3CA: phosphatidylinositol-4,5-bisphosphate 3-kinase catalytic subunit alpha, TP53: tumor protein p53, PTEN: phosphatase and tensin homolog, * Nonsense mutation, ‡ Some BRCA1 variants are reported in BT-474, UACC-812, and MDA-MB-361 cells, but they are not considered to be pathogenic and regarded as wild-type BRCA1 according to [35]. ¶ Mutation site is located outside the central DNA-binding core [47]., § Reported as wild-type in some studies [34,44,46,48].

**Table 3 cancers-12-01811-t003:** Signature Genes Associated with Poor Survival of HER2-positive Breast Cancer Patients.

Gene			HER2-Positive			METABRIC-ALL		
Official Gene Symbol	Gene Name	Loci	*p*-value	High_Mean (Month)	Low_Mean (Month)	*p*-value	High_Mean (Month)	Low_Mean (Month)
High								
CD9	CD9 molecule	12p13.31	0.00311	98.47881944	177.3011905	0.0722	123.8943511	127.9186293
UBL3	ubiquitin like 3	13q12.3	0.00853	99.91584699	151.0207207	2.35 × 10^-5^	129.3927165	121.1346243
TM4SF1	transmembrane 4 L six family member 1	3q25.1	0.00999	98.17402863	138.0701754	0.00971	126.4676768	114.0395173
MGST3	microsomal glutathione S-transferase 3	1q24.1	0.0108	99.85749559	161.2677419	0.148	123.9532677	128.1961192
NTN4	netrin 4	12q22	0.0234	94.23703703	113.1540161	0.00656	135.0298174	121.5295063
NOVA1	NOVA alternative splicing regulator 1	14q12	0.0248	100.9984848	116.0230303	0.018	126.2173607	114.8102178
ASPH	aspartate beta-hydroxylase	8q12.3	0.0302	102.4326531	158.1486111	0.117	123.0002328	131.1684322
EED	embryonic ectoderm development	11q14.2	0.0351	102.2848485	164.5439394	0.00837	125.8268839	122.2849961
USP53	ubiquitin specific peptidase 53	4q26	0.0366	94.01504425	123.8193146	0.26	121.9426163	133.6808
MED17	mediator complex subunit 17	11q21	0.0447	105.1683502	138.5924242	0.00204	126.367115	113.1962199
SMPDL3B	sphingomyelin phosphodiesterase acid like 3B	1p35.3	0.045	97.18613139	127.2032128	0.0127	125.2631605	124.8943857
Low								
TTF2	transcription termination factor 2	1p13.1	0.00408	112.5996616	73.4884058	3.05 × 10^-5^	122.2683455	132.0977528
CSTA	cystatin A	3q21.1	0.0188	112.0973333	100.8252381	0.0667	114.8275058	126.8276679
ABCC2	ATP binding cassette subfamily C member 2	10q24.2	0.0233	117.5826211	98.20582524	0.00525	123.9990103	132.8795455
HSPB8	heat shock protein family B (small) member 8	12q24.23	0.038	115.2397206	87.3081761	0.125	127.2404999	116.6156171
FLOT2	flotillin 2	17q11.2	0.039	111.9188482	86.06436781	0.0274	123.5043384	125.5109725
NLRP2	NLR family pyrin domain containing 2	19q13.42	0.0401	114.8108434	89.14382716	0.000946	115.3892634	126.5967624

High_mean: mean survival time of high expression group. Low_mean: mean survival time of low expression group.

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
