# Peer review of "Proliferative Classification of Intracranially Injected HER2-positive Breast Cancer Cell Lines"

_cancers, 2020, doi:10.3390/cancers12071811_

Round 1

Reviewer 1 Report

In this manuscript the authors evaluate the potential of 9 cell lines to proliferate in the brain microenviroment.

Below some comments for the authors:

Abstract:

"When cocultured with brain-residential glial cells, these two cell lines grew attached to the layer of glial cells, suggesting that they are adaptable to the
brain microenvironment."

Question: The other cells in Figure S3. also have grown in the layer of glial cells?  They just might have different growth rates. So should the conclusion reflect that. 

Results:

When injected into the murine brain parenchyma, UACC-893 and MDA-MB-453 cells 85 proliferated rapidly by 28 days after injection (Figure 1A). The growth rate of ZR-75-1, UACC-812, 86 MDA-MB-361, HCC-202, and HCC-1419 cells was slow (Figure 1B).

Questions:

1) Is there a difference in passages between these cell lines? Might be that some are more adapted to grow in plastic than in vivo. Have been in plastic for too long and have lost their ability to grow in vivo. But does not invalidate their ability of colonize the brain. Need to be clear they all have the ability to survive in the brain microenvironment just might take a bit longer to adjust to the new enviroment.

2) Did they grow any of the slow growth cell lines for longer? If yes could the data be presented?

3) It is interesting that cell line MDA-MB-361 originaly from a brain metastasis did showed slow growth. Would this suggests that during culturing this cell line have lost its ability?

 In figure S1B: Please color code your 9 cell lines so we can see where they sit. Also would be great to correlated the protein expression from CCLE (Figure S1) with the levels of protein in Figure2 and Figure S2. Seems to have a much lower variability on HER2 expression in the study than in the public data base.

Figure 2 seem to compress the data there is much more variability in the expression of HER2 in Figure S2. That more closely look with CCLE data.  Interestingly MDA-MB-453 seem to have one of the lower levels of HER2 expression? Can the authors speculate on that?
And the higher expressing HER2 cell line was slow growth.... would that suggest that HER2 has no role on their growth rate in the brain enviroment? Or all expressing HER2 will grow in brain the speed of their growth would depend on other genomic features not discussed in this manuscript?

 Figure 4 Section 2.2. It was not clear in the text which cell lines were used for this experiment (plastic, glial?)… when going to material methods for my surprise this is publically available data. Should this be mention in the text. And that bring other concerns, as it is well known that with different passages of the same cell line can harbour different genomic changes. So cell used in their experiment can look quite different form the ones used here. Second to try to compare expression would be the cell growing in glial cell the best place?

Figure 5: they mention 17 genes were associated with why present only 6?

Discussion:

MSG cells grew under these conditions (Figure 3), suggesting that our coculture system of tumor cells 239 and glial cells only partially mimicked the brain microenvironment.

Should read Figure S3? And this is a different conclusion than presented in the abstract?

Minor Point:

1) Title Fig 2: Each band 126 intensity in Figure S1B was quantified as a raw integrated density (RawIntDen) using ImageJ.  Should read Figure S2B?

2) Section 2.2 page 9 should reference Fig S3.  and explain what is presented there and how to compare to the 2 cell lines in Figure 3. They all seem to growth in glial cells.

Reviewer 2 Report

The topic of this manuscript is very interesting, and the authors have done a lot of work to explore the mechanism of breast cancer brain metastasis in HER2+ subtype. I would thank the authors for doing this important work which could help the breast cancer patients. However, there are some major defects in this manuscript which needs to be modified before it can be published.

Major concerns:
1) The authors have chosen 9 cell lines for the experiment, this is not a small number considering the complexity of intracranial injection. However, it's not sufficient for the correlational analysis as shown in Figure 2. As well, the differential expression results shown in Figure 4B may also be affected by the small sample size and need to be discussed more carefully.

2) Considering the limited number of cell lines tested, validation study in real patients is a must. And the authors have chosen the METABRIC dataset for validation. However, they didn't compare the brain metastasis rate of HER2+ and HER2- patients directly, instead they performed survival analysis. To convince the readers that the brain metastasis-signature genes identified from cell line experiments were also involved in real patients, a direct comparison of brain metastasis rate of HER2+ and HER2- patients is a must. Also, I recommend the authors to use another dataset such as TCGA to further confirm these genes.

Minor Concern:
What is the difference between P-HER2 (Y1221/1222) and P-HER2 (Y1248), and why are the last two figures in Figure 2B so similar to each other?

Round 2

Reviewer 1 Report

No further comments

Reviewer 2 Report

The authors have addressed all my concerns, no further comments.